# TRULY GENERATIVE DATA AUGMENTATION FOR IMAGE SEGMENTATION - CASE OF CLOUD IMAGES

**Mayank Jain & Soumyabrata Dev**
The ADAPT SFI Research Centre
School of Computer Science
University College Dublin, Ireland
{mayank.jain,soumyabrata.dev}@adaptcentre.ie

## ABSTRACT

Supervised learning frameworks frequently rely on semantic image segmentation, which necessitates a substantial amount of annotated data. Existing methodologies for data augmentation either employ image transformations that are limited by the cardinality of the original dataset or employ generative augmentation techniques that introduce pixel categorization errors. This paper presents an innovative approach for "truly" generative data augmentation for image segmentation, specifically in the context of sky/cloud images. The proposed method involves separate generation of the background clear sky image and the foreground cloud masks using two separate DCGANs, which are subsequently merged to produce augmented images. This organic approach enhances the quality of generated images while preserving accurate pixel categorization. The proposed approach is finally noted to improve the robustness of the sky/cloud image segmentation models.

## 1 INTRODUCTION

Semantic image segmentation (SIS) plays a vital role in understanding the contents of an image as it assigns a class or object to each pixel. However, the SIS annotation process is accurate but expensive, resulting in limited availability of annotated images, especially for small and specific applications Liu et al. (2019). To address this limitation, researchers rely on standard data augmentation strategies (DAS) Dev et al. (2019). Standard DAS techniques involve transforming images through rotation, flipping, and altering colour schemes. Although these techniques improve the diversity of semantically segmented images, they are limited by the size of the original dataset and may reduce the performance and generalisability of trained models when applied in excess Cubuk et al. (2019). To overcome these challenges, several researchers have tried to use generative models to augment SIS datasets Bowles et al. (2018); Sandfort et al. (2019); Jain et al. (2021). These models generate diverse images that lie within the distribution of the data and hence can be further subjected to standard DAS. However, when it comes to the generation of the ground truth segmentation map (GTSM), they effectively perform the segmentation either intrinsically or explicitly, raising concerns about the adoption of the entire process to augment the SIS dataset and train another segmentation model. Accordingly, this paper proposes a novel and organic method for augmenting SIS datasets in the case of sky/cloud image segmentation without explicitly performing any sort of semantic segmentation at all.

## 2 PROPOSED APPROACH - CASE OF SKY/CLOUD IMAGE SEGMENTATION

Identifying cloud cover by performing SIS on sky/cloud images is crucial for many meteorological studies Dev et al. (2019). Our paper formulizes the problem as a human artist who is asked to perform the augmentation task. Organically, the first step would be to paint a clear sky layer and then add clouds. However, while painting the clouds, the artist can place a tracing paper to get an impression of the annotation mask. Accordingly, we propose to train two separate deep convolutional generative adversarial networks (DCGAN) Radford et al. (2016), one for the background clear sky layer (say, SkyGAN) and the other for the clouds to be added in the foreground (say, CloudGAN). SkyGAN was trained on 224 clear-sky images available in SWIMCAT Dev et al. (2015). For sky/cloud

image segmentation datasets, this paper uses two datasets, namely SWIMSEG which contains 1013 annotated sky/cloud images-GTSM pairs Dev et al. (2016), and HYTA which contains only 32 such pairs. Hence, CloudGAN was trained separately on both the train split of the SWIMSEG dataset and the complete HYTA dataset (where no train-test split was made due to the small cardinality). The generated clear-sky images and the cloud mask foreground (CMF) are shown in Figure 1 (a) and Figure 1 (b,c), respectively. Note that the black portion of the cloud mask is enough to directly obtain the GTSM mask, as shown in Figure 1 (e). The sky/cloud image is obtained by merging the cloud mask foreground image with the generated clear-sky background. This is done by considering black pixels as transparent pixels. Consequently, the proposed approach not only is truly generative in nature, but also removes any likelihood of error in estimating GTSM for the generated sky/cloud image.

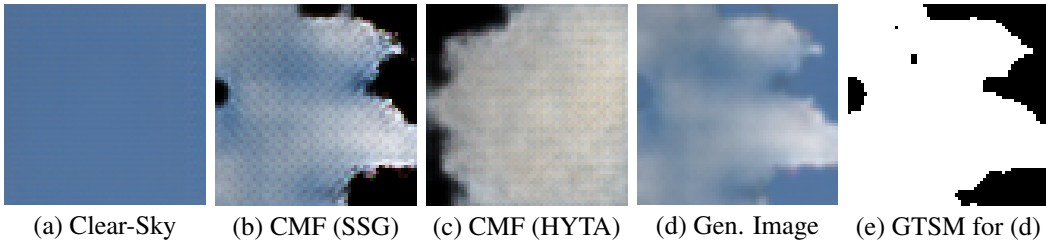

|          |          |          |          |          |
| :------: | :------: | :------: | :------: | :------: |
| (a) Clear-Sky | (b) CMF (SSG) | (c) CMF (HYTA) | (d) Gen. Image | (e) GTSM for (d) |

Figure 1: (a) Sample clear-sky background image generated by SkyGAN; (b,c) Sample cloud mask foregrounds generated by CloudGAN when trained on SWIMSEG (train split) and HYTA dataset respectively; (d) Complete sky/cloud image obtained by merging (b) with (a); (e) Ground truth segmentation mask obtained from (b) to correspond to the complete sky/cloud image (d).

To evaluate the effectiveness of the proposed approach, CloudSegNet Dev et al. (2019) is used as the sky/cloud SIS model. The architecture was specifically proposed in the literature to perform sky/cloud SIS. The segmentation model was trained three times to report results for three augmentation policies, namely (a) NoAug - where no augmentation is performed, (b) StdAug - where standard augmentation policies, *i.e.* random rotations and flips, were performed, and (c) GenAug - where 300 images are augmented using the proposed generative approach. Since generative augmentations always allow the use of standard augmentation policies, StdAug augmentation policies are also used in the case of GenAug. Furthermore, all experiments were conducted separately on the SWIMSEG dataset and the HYTA dataset. For the HYTA dataset, in the absence of a test set, the models are only evaluated on the out-of-distribution (OOD) SWIMSEG dataset, whereas in the other case, performance is evaluated on both in-distribution test set and OOD HYTA dataset.

While performance of GenAug approach slightly decreases ($\sim 1.5\%$ accuracy) on the in-distribution test set in the case of SWIMSEG dataset, the results were superior on the OOD dataset ($\sim 3.5\%$ accuracy) for both cases showing that the proposed augmentation approach improves generalizability of the SIS model.[1]

## 3   CONCLUSION AND FUTURE WORK

This paper introduces a truly generative approach augmenting cloud/sky image segmentation datasets. The proposed method ensures accurate matching of generated sky/cloud images with the corresponding binary segmentation maps by training separate DCGANs for generating clouds for the foreground and clear-sky image for the background. While this paper demonstrates a definite improvement in the robustness and the generalizability of the segmentation models by employing the proposed approach, further improvements are needed to enhance its applicability to a wider range of image segmentation problems. Future directions also include exploring the context exchange between the two DCGANs to achieve coherent generation of blue shades in the cloud masks and background clear sky images. With exploration of these factors, the proposed method holds potential to advance the field of data augmentation for image segmentation while improving the overall quality of generated image-annotation pairs.

---

[1]Refer to the appendix A.1 for the complete performance evaluation table.

ACKNOWLEDGEMENTS

This research was conducted with the financial support of Science Foundation Ireland under Grant Agreement No. 13/RC/2106_P2 at the ADAPT SFI Research Centre at University College Dublin. ADAPT, the SFI Research Centre for AI-Driven Digital Content Technology, is funded by Science Foundation Ireland through the SFI Research Centres Programme.

URM STATEMENT

The authors acknowledge that at least one key author of this work meets the URM criteria of the ICLR 2023 Tiny Papers Track.

CODE RELEASE

For reproducibility, the codebase used for the experiments carried out in this article is released at `https://github.com/jain15mayank/Truly-Generative-Augmentation-CloudSeg`.

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

# A   APPENDIX

## A.1   RESULTS OF SEGMENTATION EXPERIMENTS

| Train On | Methods | Test Set | | | Out-Of-Distribution Dataset | | |
|---|---|---|---|---|---|---|---|
| | | Accuracy | mIoU | MCC | Accuracy | mIoU | MCC |
| SWIMSEG (train) | NoAug | **87.50**% | **0.7752** | **0.7465** | 70.26% | 0.5414 | 0.4306 |
| | StdAug | 86.91% | 0.7660 | 0.7348 | 71.08% | 0.5512 | 0.4433 |
| | GenAug | 85.91% | 0.7475 | 0.7101 | **75.16**% | **0.6012** | **0.5065** |
| HYTA (complete) | NoAug | - | - | - | 72.77% | 0.5912 | 0.4832 |
| | StdAug | - | - | - | 71.30% | 0.5838 | 0.5250 |
| | GenAug | - | - | - | **76.28**% | **0.6261** | **0.5583** |

Table 1: Results as obtained for training the CloudSegNet Dev et al. (2019) on the SWIMSEG and HYTA datasets. Since HYTA is a small dataset, with a cardinality of only 32 image segmentation map pairs, all were used for training and therefore there was no within-distribution test set to evaluate the trained models on. The models were trained three times on (a) NoAug - data without any augmentation, (b) StdAug - data with standard augmentation policies, *i.e.* random rotations and flips, and (c) GenAug - data with both generative augmentation by 300 images and StdAug. Since generative augmentations always allow the use of standard augmentation policies, the results are reported only for GenAug with StdAug. The metrics used for the comparison are 'accuracy'$\in [0, 100]$, 'mean intersection over union (mIoU)'$\in [0, 1]$ and 'Matthew's correlation coefficie (MCC)'$\in [-1, 1]$. Superior results are marked in bold for each experiment and evaluation data.

## A.2   SWIMSEG GENERATED IMAGES

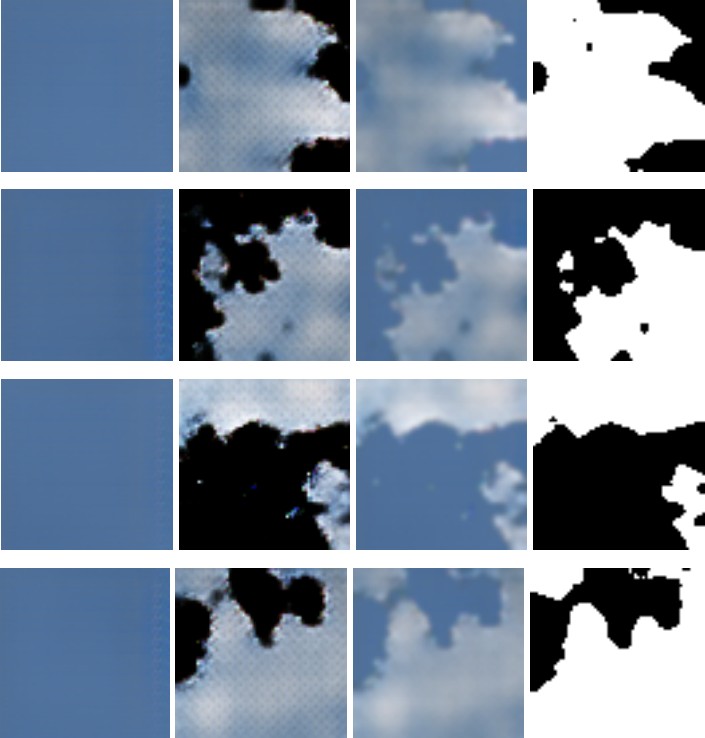

Figure 2: Sample Images Generated for SWIMSEG data - clear sky images from SWIMCAT data is in the extreme left. It is followed by randomly selected "only cloud" images extracted from the SWIMSEG test data. The first two images are then merged to obtain the third image in each row. The fourth image of each row reflects segmentation map of second image.

## A.3 HYTA GENERATED IMAGES

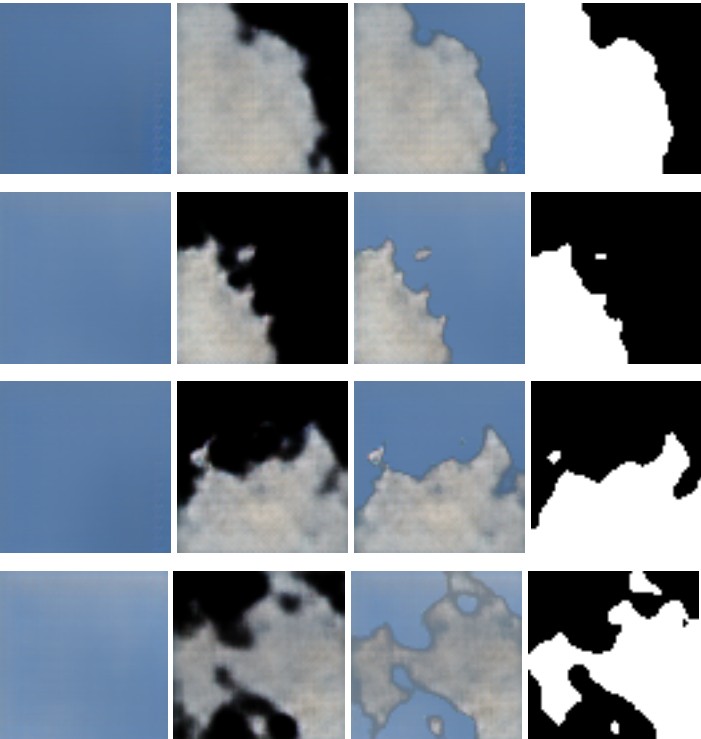

Figure 3: Sample Image Generated in HYTA data - clear sky images from SWIMCAT data is in the extreme left. It is followed by randomly selected "only cloud" images extracted from HYTA dataset. The first two images are then merged to obtain the third image in each row. The fourth image of each row reflects segmentation map of second image.

