# OpenReview forum: "Truly Generative Data Augmentation for Image Segmentation - Case of Cloud Images"
_ICLR.cc/2023/TinyPapers — Submitted to Tiny Papers @ ICLR 2023_

### Official Review · Reviewer_uvVC · 2023-03-31

**Confidence:** 3

**Summary Of Contributions:**

This work proposed a generative data augmentation method for image segmentation. Two GANs are designed to generate and compose images for sky-cloud image segmentation.

**Rating:**

Great Start (GS): a submission which meets some of the reviewing criteria but has room for improvement

**Strengths And Weaknesses:**

### Strengths

1. Clear writing. This paper provides a well-organized context for reviewers to understand the generative data augmentation problem. Methods are properly motivated.

2. Visually compelling results. Generated data augmentations are visually close to the samples from the original dataset.

### Weaknesses

1. Comparison. Given the purpose of this work is to augment resource-constrained image segmentation task, it could make this work more convincing to train a small model over the original data and the joint of the original dataset and augmented dataset.

2. Paper organization. The writing is overall clear, but the paper can be organized better. Authors are expected to streamline the introduction section and give more attention to the method description.

**Suggested Changes:**

Authors are encouraged to improve this work from these aspects.

1. Include a comparison of the performance of a model trained on the augmented data vs. a model trained on the original data, to demonstrate the effectiveness of the proposed method for resource-constrained image segmentation.

2. Streamline the introduction section better to highlight the motivation and contribution of the proposed method, and provide more detail and clarity in the method description section.

---

### Meta-Review · Area_Chair_E2G4 · 2023-04-08

**Recommendation:** Invite to revise
**Confidence:** 5

**Metareview:**

The authors propose a novel approach to data augmentation for image segmentation by using a pair of GANs to make this a fully generative procedure rather than, for example, relying on existing segmentation methods as some previous approaches do. This is an interesting approach and it would be great if the authors could spend more time diving deeper into a description and evaluation of their exciting proposed method. As the reviewer mentioned, this can be done by cutting down on the introduction, expanding the discussion, and adding some comparisons to existing methods.

**Summary:**

The authors propose using a pair of GANs to augment data for image segmentation. Providing more details and evaluations of this novel method would be very helpful.

**Comments And Feedback To The Authors:**

Please see the suggestions above, but to summarize, even though space is limited, it would be great to better highlight your novel method and compare it against existing methods. Shortening the introduction and including an appendix could help provide the space necessary to do this.

**Reason For Not Giving A Higher Recommendation:**

More discussion of the details of the proposed method and some comparisons to existing methods would greatly strengthen the clarity and reproducibility of this submission. This would in turn help with evaluation of the correctness of the proposal.

**Reason For Not Giving A Lower Recommendation:**

N/A

---

### Meta-Review · Area_Chair_ua4a · 2023-04-08

**Recommendation:** Invite to revise
**Confidence:** 4

**Metareview:**

The paper proposes a generative data augmentation method for image segmentation by training two separate GANs to generate foreground cloud masks and background clear sky images, respectively, and then merging them together. The paper is well-written and provides visually compelling results. However, the paper could be organized better, and the proposed method could benefit from a comparison of its performance against a model trained solely on the original data.

**Summary:**

The paper proposes a novel generative data augmentation method for image segmentation using two separate GANs to generate foreground cloud masks and background clear sky images. The generated images are then merged together to create new training data for image segmentation models.

**Reason For Not Giving A Higher Recommendation:**

The paper lacks a comparison of the performance of a model trained on the augmented data vs. a model trained on the original data to demonstrate the effectiveness of the proposed method. Authors are encouraged to train a tiny model over the augmented dataset to verify the utility of proposed generative data augmentation and also provide quantitative evaluation using the training results.

Additionally, the paper could be better organized, with a streamlined introduction section and more detail and clarity in the method description section.

**Reason For Not Giving A Lower Recommendation:**

N/A

---

### Decision · Program_Chairs · 2023-04-09

Revision accepted; invite to archive